# Adaptive Network Model for Assisting People with Disabilities through Crowd Monitoring and Control

**DOI:** 10.3390/bioengineering11030283

**Published:** 2024-03-16

**Authors:** Alicia Falcon-Caro, Evtim Peytchev, Saeid Sanei

**Affiliations:** 1Department of Computer Science, Nottingham Trent University, Nottingham NG11 8NS, UK; evtim.peytchev@ntu.ac.uk (E.P.); s.sanei@imperial.ac.uk (S.S.); 2Department of Electrical and Electronic Engineering, Imperial College London, London SW7 2AZ, UK

**Keywords:** adaptive networks, assistive technologies, AirTag, crowd monitoring, disability, pandemic, tracking devices

## Abstract

Here, we present an effective application of adaptive cooperative networks, namely assisting disables in navigating in a crowd in a pandemic or emergency situation. To achieve this, we model crowd movement and introduce a cooperative learning approach to enable cooperation and self-organization of the crowd members with impaired health or on wheelchairs to ensure their safe movement in the crowd. Here, it is assumed that the movement path and the varying locations of the other crowd members can be estimated by each agent. Therefore, the network nodes (agents) should continuously reorganize themselves by varying their speeds and distances from each other, from the surrounding walls, and from obstacles within a predefined limit. It is also demonstrated how the available wireless trackers such as AirTags can be used for this purpose. The model effectiveness is examined with respect to the real-time changes in environmental parameters and its efficacy is verified.

## 1. Introduction

Most people require a kind of assistive technology at some point in their lives, especially as they age or face disability. While some may require assistive technology temporarily, such as after an accident or illness, others may require it for a longer period or throughout their lifespan. This technology is most needed by older people, children and adults with disabilities, and people with long-term health conditions such as diabetes, stroke, and dementia. Assistive products can range from physical products such as wheelchairs, prosthetic limbs, and hearing aids to digital solutions such as speech recognition or automated safeguarding. Improving access to assistive technology can contribute to the achievement of the sustainable development goals and ensure that no one is left unattended. Such a technology is developed and deployed in many ways. In this paper, advanced adaptive signal processing is used to develop a wireless decentralized multi-agent communication network to assist and protect people with disabilities in a crowd during an emergency situation, such as a pandemic.

COVID-19 deeply affected the world in the past five years. During the pandemic, concern about how a crowd of people moves and how the people interact increased [1]. Therefore, work on crowd monitoring and analysis, and how to intervene in the crowd structure, attracted the attention of more researchers. Although some new advances in crowd monitoring have been made to tackle this problem and to maintain a safe social distance, to the best of our knowledge, none of these methods has focused on a unified and inclusive adaptive network analysis approach which also caters for people with disabilities, such as blind or visually impaired people as well as wheelchair users.

Most recently proposed crowd monitoring techniques are fusion-based (centrally controlled) and rely on the use of surveillance cameras powered by image processing and computer vision algorithms [2]. These methods are passive, have enormous technical and privacy limitations, and do not allow prediction and easy intervention of the movement and behavior of the crowd. Even so, some of the latest crowd analysis methods are now starting to use spatial–temporal data and apply some advanced feature learning and classification methods such as recurrent neural networks (RNNs) and deep neural networks (DNNs). These approaches enable the prediction of crowd behavior and its direction of movement [3]. Yet, the crowd elements (people) do not benefit from interactions with their neighborhoods and local interventions. Therefore, these methods are not useful enough for assisting people with disabilities navigate in a crowd.

Most advances in visual assistive technologies [4] rely on computer vision techniques and the use of a global positioning system (GPS) [5]. The centralized vision-based systems or those using GPS share limitations and problems with the previously mentioned crowd monitoring systems. The challenges and limitations are related to low signal strength, sharing personal information, problems with both indoor and outdoor operations, and also low location accuracy [6,7]. By indoors, we refer to places with weak or no internet connection. Some of these technologies also require the users to wear some intrusive devices, making users more reliant on such wearables. On the other hand, for wheelchair users, although some advances have been made in autonomous wheelchair design [8,9], these technologies often rely on the use of computer vision and other technologies which are subject to some ethical and privacy regulations. These systems often need to learn the environmental map before they can be used with confidence. This makes the use of such systems difficult for a constantly varying environment.

Facing these challenges, we develop a comfortable-to-carry and easy-to-use decentralized system equipped with low-range communication tools for assisting people with disabilities in moving through a crowd conveniently while maintaining a safe distance from other people or obstacles. The preliminary results of this work have been presented in [10]. The proposed system models crowd movement in a dynamic environment and a distributed manner depending on the information the agents receive from each other and the changes in the environment. This allows for tracking the agents (people) including people with disabilities who share a connected network. The environment may include well-defined constraints, such as walls and fences, ticket control barriers, objects, or people moving in unpredictable directions. To perform this analysis, we use the concept of adaptive cooperative networks by means of the diffusion adaptation mechanism [11,12] to model the crowd motion while passing through geometrically varying areas. The diffusion adaptation over networks strategy was chosen for this application due to its successful results in network modeling and swarming, including biological networks, such as bird flight formation [13] or fish schools [14], and its promising results in modeling and monitoring a crowd of people [15]. Therefore, in our proposed method, this adaptive technique replaces traditional distributed systems which require exhaustive programming or solving cumbersome differential equations.

In this work, the agents (or nodes representing people) share their position coordinates and each agent communicates with other agents within its one-hop neighborhood. As long as an agent can detect the positions of the other nodes (agents) in its neighborhood, it adjusts its speed and distance with others and the barriers while moving towards its destination (e.g., the exit gate in a metro station). This helps in more accurately calculating and maintaining safe distances between the general public and people with disabilities as well as safe social distances in pandemic situations. To achieve this, the movement speed, the distance between the agents, and possibly their movement directions must change (within allowed limits) with respect to the variations in the pathway geometry (e.g., width) and any obstacle preventing them in reaching their target. This can be achieved simply by being aware of the nodes within a neighborhood and the geometrical constraints. In this scenario, people can make a compromise between their distances and speeds, to keep themselves safe.

The proposed system must also work with considerably small variations in distances between the agents in the range of centimeters, to be able to operate both indoors and outdoors, including underground (e.g., metro corridor). A number of positioning technologies can be used to obtain the agents’ location in real-time, some of which are more appropriate than others for the proposed scenario. For instance, a geolocation technology such as GPS is widely used and easily available. However, it usually has an accuracy of 4 to 5 m and cannot operate underground, which makes it inappropriate for this application. The latest Bluetooth technologies have a high location accuracy in the centimeter range and can work quite well indoors and outdoors [16,17,18]. However, it may have a low signal strength for larger neighborhoods, which can cause loss or delays. The above limitations make such systems inadequate for decentralized crowd monitoring on their own, especially underground or places with no internet access. Therefore, we propose the use of tracking devices with embedded short-range wireless communication systems, such as ultra-wideband (UWB) [19], which can provide a high-precision positioning within the centimeter range. An example of these tracking devices is the AirTag from Apple Inc., compatible with Apple devices, or AirFinders from Link Labs, compatible with most modern smartphone operating systems (OSs). These devices, however, need to be linked to a compatible mobile device, making them insufficiently practical. To avoid this problem, we could instead use the Precision Finding feature available in the most recently released Apple devices, which makes use of the embedded UWB technology to precisely locate other compatible Apple devices in its neighborhood.

To use this system for crowd monitoring, each agent of the network, representing a member of the crowd including a person with disabilities, needs to carry one of these smart devices. Hence, the location of the device and the subject’s movement direction can be accessed in real-time and fed onto the diffusion adaptation model as the agents’ position coordinates.

The main contributions of this paper are as follows: (1) development of a diffusion adaptation model which incorporates the environmental parameters into its formulation; (2) development of an assistive technology based on cooperative networking that can help people with disabilities navigate a crowd in a challenging environment; and (3) using state-of-the-art commercially available smart devices for decentralized crowd monitoring purposes.

## 2. Materials and Methods

### 2.1. Diffusion Adaptation Modeling

The concept of diffusion adaptation cooperative networks has opened a new direction in adaptive and distributed signal processing and analysis of multi-agent communication networks [11,12]. These networks have the capability of modeling groups of nodes or agents which can transfer information to each other and try to achieve common target(s) in a cooperative manner. For more than one target or objective, multi-task scenarios [20] have also been attempted and used for biological network modeling [13,14,21] and in social networks [22] as very common applications. The same adaptive systems can be used to check the reliability of the received information in a multi-agent computer network. This is useful in maintaining the security of such networks [23]. A wider application of the technique can be seen in medicine where the medical images are classified through a cooperative dictionary learning approach [24]; or, considering the electrodes of an electroencephalography system as the nodes of a cooperative network, such a system can find applications in brain–computer interfacing [25,26].

Here, to model a crowd moving through a geometrically varying environment over time, we introduce a mobility model of people, represented as nodes or agents of a connected network, using diffusion adaptation.

#### 2.1.1. Diffusion Adaptation

In multi-agent distributed networks, the agents collaborate with each other to solve a global optimization problem. In particular, for crowd modeling, each agent *k* is interested in estimating an unknown target or objective τ, while sharing information with the agents in its neighborhood $N_{k}$. As one of the distributed learning strategies, diffusion adaptation is a symmetric and stable consensus strategy defined in [11] and further developed by many researchers, some referred to in Section 2.1. The algorithm has two parts of adaptation and combination which can alternate between different orders: Adapt-Then-Combine (ATC) or Combine-Then-Adapt (CTA). The adaptation convergence of multi-agent diffusion networks has been proved in the literature [12].

Consider the crowd as a collection of people distributed over a space ${ℜ}^{2}$ with a defined geometry. The collection of people, with the ability to communicate to each other and share information, forms an adaptive network. They adapt their movement to those of agents in the neighborhood as well as geometrical/spatial constraints while moving towards their target which, in this example, is at the end of the predefined path (e.g., an exit door). This also helps the crowd members to self-organize themselves based on the information exchanged within their one-hop neighbors. Figure 1 illustrates a group of agents, their neighborhood, and an exemplar of the surrounding environment.

The general objective of such a network is for each node *k* to reach the location of the target in a fully distributed manner. One option to achieve this objective is to use the ATC diffusion algorithm [14,27,28].

The model presented here is therefore a variation of the models studied in [10,14,15]. This variation allows the network to move towards a target smoothly through a predefined path while avoiding possible obstacles. Following the diffusion adaptation strategy given in [12], consider a connected network of *N* nodes where each node *k* wants to estimate an unknown parameter τ from the collected local measurements $\left\{d_{k,i},u_{k,i},w_{k,i}\right\}$ for each node *k* at time instant *i*. This is achieved through the estimation of the global parameter τ that minimizes the cost function given by
(1)$$J^{global}\left(\tau \right)=\sum_{k=1}^{N}E{\left|d_{k,i}-u_{k,i}\tau \right|}^{2}$$
where E is the expectation operator. $u_{k,i}$ represents a unit direction regression vector pointing to the direction of the target. $w_{k,i}$ is the location vector of node *k* relative to a global coordinate system at each time instant *i* and is discussed in Section 2.1.2. $d_{k,i}$ represents the scalar distance between the location of the target and $w_{k,i}$, and is given by the inner product:(2)$$d_{k,i}=u_{k,i}\left(\tau -w_{k,i}\right)$$
To solve the optimization problem in (Equation 1), we use the ATC diffusion strategy, which gives the following set of distributed adaptive equations:(3)$$\psi_{k,i}=\tau_{k,i-1}+\mu_{k}u_{k,i}^{T}\left[d_{k,i}-u_{k,i}\tau_{k,i-1}\right]$$
(4)$$\tau_{k,i}=\sum_{l\in N_{k}}a_{l,k}\psi_{l,i}$$
where $u_{k,i}^{T}$ represents the transpose of $u_{k,i}$, $N_{k}$ is the number of agents in the neighborhood of node *k*, and $\mu_{k}$ is a positive step size used by node *k*. The combination weight representing the information received from each node *l* ($a_{l,k}$) is from a set of non-negative real weights assigned to node *k* and satisfies
(5)$$\sum_{l=1}^{N}a_{l,k}=1,\;\;\;a_{l,k}=0\;\;if\;l\notin N_{k}$$

In the implementation of Equations (Equation 3) and (Equation 4), the nodes in the node *k* neighborhood share their intermediate estimates {$\psi_{l,i},d_{k,i},u_{k,i}$} after each iteration. Since our model is geometrically bearing, we can simplify Equation (Equation 3) under reasonable approximations to [28]
(6)$$\tau_{k,i-1}-\tau_{k,i}=\left|\right|\tau_{k,i-1}-\tau_{k,i}\left|\right|u_{k,i}^{T}$$
Hence, the equation can finally be described as
(7)$$\psi_{k,i}=\left(1-\mu_{k}\right)\tau_{k,i-1}+\mu_{k}\tau_{k,i}$$

#### 2.1.2. Motion Model

Similar to mobile networks, in our crowd movement scenario, the relationship between the movement speed and two consecutive agent locations is defined as [14,28]
(8)$$w_{k,i+1}=w_{k,i}+\Delta i\cdot v_{k,i+1}$$
where Δi is the time step (time difference between two consecutive states) and $v_{k,i+1}$ is the velocity vector of node *k* in the next time instant i+1. Therefore, from now on, we focus on estimating the velocity vector and the current position of each agent *k* at each time instant *i*, denoted by $w_{k,i}$.

In our model, there are two factors that influence the velocity vector of the nodes. The first factor is the spatial constraint involved in identification of the location of node *k* at each time instant *i*. In our model, we want the crowd to navigate through a predefined path from a start point to the end point. In Figure 1, we see how the distance between the two surrounding walls of the pathway can change. The moving direction for the crowd (from left to right) is denoted by an arrow. In such a scenario, while the safe social distancing is followed, in the wider areas, the people can walk normally and have moderate to large distances between them. Nevertheless, in the narrower regions, the subjects should move faster while allowing a minimum, smaller (yet permitted) social distancing to avoid a traffic jam in narrow areas.

The objective of our model is to estimate the position of agent $w_{k,i}$, while the agent moves between the two walls and keeps its permissible distance limit from other agents.

The second factor that influences the velocity vector of the nodes is the desire of the agents to move in synchrony and avoid collisions by maintaining a safe distance *r* between the nodes. As described in [28], this can be achieved by updating the velocity vector as follows:(9)$$v_{k,i+1}^{b}=v_{k,i}^{g}+\gamma \delta_{k,i}$$
where γ is a non-negative scalar and $\delta_{k,i}$ is given by
(10)$$\delta_{k,i}=\frac{1}{|N_{k}|-1}\sum_{l\in N_{k}\backslash \left\{k\right\}}\left(∥w_{l,i}-w_{k,i}∥-r\right)u_{k,i}\left(w_{l,i}-w_{k,i}\right)$$

$v_{k,i}^{g}$ refers to a local estimate for the velocity of the center of gravity of the network and is obtained by the ATC diffusion strategy described in (Equation 1)–(Equation 7) as
(11)$$\varphi_{l,i}=\left(1-\mu_{k}^{v}\right)v_{k,i-1}^{g}+\mu_{k}^{v}v_{k,i}$$
(12)$$v_{k,i}^{g}=\sum_{l\in N_{k}}a_{l,k}^{v}\varphi_{l,i}$$
where $\mu_{k}^{v}$ is a positive step size, and $a_{l,k}^{v}$ are the combination weights which satisfy similar conditions to those in (Equation 5).

#### 2.1.3. Motion Model with Variable Speed and Distance between Nodes

To model a more realistic crowd motion, Equations (Equation 8)–(Equation 12) are modified so the speed of each node *k* and the distance between the nodes can be scaled depending on the width of the crowd pathway where node *k* lands at time instant *i* and the distances between individual nodes and the target (effectively for the narrow regions).

In order to predict (or estimate) the new speed and location for agent *k*, we need to re-calculate the above parameters based on the closeness of the two surrounding walls. In our simulation, we assume that the agents move inside a region restricted by two walls, where the dimensions are approximated by the chords of circles tangent to both walls at time instant *i*. The chord links the two tangent points. The position of node *k* is considered to be on the corresponding chord (i.e., the chord where *k* falls on). Figure 2 clearly shows the concept. The agents also maintain a minimum predefined safe distance from the walls.

To enable a more realistic scenario, we assume that the people often go as slow as one step and as fast as three steps per second with approximately 0.6 to 1.2 m strides, respectively. This assumption is essential for setting the initial and the baseline crowd speed. This means that the speed of node *k* at time instant *i* ($v_{k,i}$) can vary between $v_{min}=0.6$ m/s and $v_{max}=3.6$ m/s. This gives an average speed of $v_{avg}=2.1$ m/s and can be assumed fixed for all the agents representing the general public. Agents representing the individuals with disabilities are assumed to have half of the normal speed. Given their physical condition, it is reasonable to assume that they always move at a slower speed than individuals without disabilities. The same assumption could be made for toddlers and older people if we were to include them as part of the simulated crowd.

On the other hand, the minimum social distance *r* can also vary inversely proportional to the speed (or according to the closeness of the walls) between a lowest (e.g., $r_{min}=1$ m) and a highest (e.g., $r_{max}=2$ m) value. For people with disabilities, the minimum social distance is higher than for those of the general public given the same speed.

Now, the objective of the new model is to allow the nodes to have a higher speed and a reasonably lower distance between the nodes for the narrower regions (closer walls) and vice versa. Based on this assumption, the effective social distance in the neighborhood of node *k* at time instant *i* can be defined as
(13)$$r_{k,i}=\frac{v_{max}-\left(v_{k,i}\cdot \frac{t_{k,i}-w_{k,i}}{∥t_{k,i}-w_{k,i}∥}\right)}{v_{max}-v_{min}}\left(r_{max}-r_{min}\right)+r_{min}$$
and can be numerically approximated using the previously defined $v_{max}$, $v_{min}$, $r_{max}$, and $r_{min}$ as
(14)$$r_{k,i}=\frac{3.6-\left(v_{k,i}\cdot \frac{t_{k,i}-w_{k,i}}{∥t_{k,i}-w_{k,i}∥}\right)}{3}+1$$
where $t_{k,i}$ represents the target (or end point) location vector at each time instant *i*.

This shows a linear (but negative) dependency between the social distancing of agent *k* and its speed at time instant *i*. Therefore, as long as the speed is known by the agent, the social distance can be estimated instantly. For individuals with disabilities, their social distance is expected to be
(15)$$r_{k,i}^{disabled}=2r_{k,i}$$

To estimate the speed, we refer to Figure 2. At each time instant *i*, agent *k* falls on the chord of a circle linking the two tangent points between the walls and the circle. This is a unique chord for each agent. The agent’s speed is inversely proportional to the corresponding chord length at that time.

To calculate such a chord, knowing the two functions that represent the walls and the coordinates of node *k* at time instant *i* relative to the same global coordinate system, as shown in Figure 2, the centers of all the circles fall on the center dashed line. Therefore, the chord equation can be defined as
(16)$$\frac{y-y_{k}^{1}}{y_{k}^{2}-y_{k}^{1}}=\frac{x-x_{k}^{1}}{x_{k}^{2}-x_{k}^{1}}$$
where $\left(x_{k}^{1},y_{k}^{1}\right)$ and $\left(x_{k}^{2},y_{k}^{2}\right)$ are the tangent points of the two walls and the circle, as well as two points of the desired circle centered at $O_{k}$. Therefore, the circle can be defined by the following equations:$${\left(x_{k}^{1}-x_{O_{k}}\right)}^{2}+{\left(y_{k}^{1}-y_{O_{k}}\right)}^{2}=r_{O_{k}}^{2}$$
(17)$${\left(x_{k}^{2}-x_{O_{k}}\right)}^{2}+{\left(y_{k}^{2}-y_{O_{k}}\right)}^{2}=r_{O_{k}}^{2}$$
where $x_{O_{k}}$ and $y_{O_{k}}$ represent the coordinates (x,y) of the circle center $O_{k}$, and $r_{O_{k}}$ is the radius of such a circle tangent to the two walls, which is also the distance between the walls and the center line. In (Equation 16) and (Equation 17), we drop the time index *i* for simplicity.

To obtain all the necessary variables in the above equations and make the chord length measurable, assume $f_{1}\left(x\right)$ and $f_{2}\left(x\right)$ are the known equations for the two walls. Therefore, another two equations of our system of equations are
$$y_{k}^{1}=f_{1}\left(x_{k}^{1}\right)$$
(18)$$y_{k}^{2}=f_{2}\left(x_{k}^{2}\right)$$
In this case, the chord length of node *k* at time instant *i*, which is the main parameter for geometrical adaptation, is calculated as
(19)$$L_{k,i}={\left({\left(y_{k}^{2}-y_{k}^{1}\right)}^{2}+{\left(x_{k}^{2}-x_{k}^{1}\right)}^{2}\right)}^{1/2}$$

Finally, we need to utilize the information about the maximum and minimum widths of the pathway, which represent the maximum and minimum chord lengths, respectively, to adjust the agent speed and, accordingly, social distance. Given these two values of $L_{min}$ and $L_{max}$, and associating them, respectively, to $v_{max}$ and $v_{min}$, the speed factor $v_{k,i}^{c}$ can be approximated as
(20)$$v_{k,i}^{c}=\frac{L_{max}-L_{k,i}}{L_{max}-L_{min}}\left(v_{max}-v_{min}\right)+v_{min}$$
By replacing the parameters from (Equation 20) with realistic values of $v_{min}=0.6$ m/s and $v_{max}=3.6$ m/s and to proceed with some experiments, $v_{k,i}^{c}$ is approximated as
(21)$$v_{k,i}^{c}=3\cdot \frac{L_{max}-L_{k,i}}{L_{max}-L_{min}}+0.6$$

In practice, for large crowds, we may assume that there is no chord (pathway width) of less than 1 m in width and no need for any concern about social distancing for pathways of more than 10 m in width (chord length). In that case:(22)$$v_{k,i}^{c}=3\cdot \frac{10-L_{k,i}}{9}+0.6$$
Hence, by measuring the chord of agent *k* at time instant *i*$L_{k,i}$, we are able to find all other parameters for a cooperative movement of the agents towards their common destination.

The new estimated speed is applied to the overall velocity vector of node *k* in order to scale and adjust the speed depending on the cross-section of the area where the node is located at time instant *i*. To ensure that the system works if the crowd interacts with an obstacle, the new velocity vector is set as follows:vk,i+1a=(23a)Ck,itk,i−wk,i∥tk,i−wk,i∥(23b)−Ck,iR∥wk,i−(pk,i−α)∥−1wk,i−(pk,i−α)∥wk,i−(pk,i−α)∥
where *R* is the radius of the node *k* neighborhood, $p_{k,i}$ represents the location vector in the coordinates (x,y) of the obstacle that node *k* wants to avoid at time instant *i*, and α is an adjustable coefficient that allows the node to maintain a safe distance from such an obstacle. $C_{k,i}$ is a coefficient that regulates the speed of each node *k* at each time instant *i* depending on the path width:(24)$$C_{k,i}=v_{k,i}^{c}$$
Therefore, when node *k* does not face any static obstacle such as a barrier or walls, $v_{k,i+1}^{a}$ follows (23a) and moves towards the target $t_{k,i}$. On the other hand, when node *k* detects an obstacle close to its location, it avoids the obstacle and moves in the direction opposite to the obstacle (23b).

According to (9)–(23), we propose the following mechanism by which $v_{k,i+1}$ can be set for node *k*. This mechanism is a modification and extension of the one proposed in [14] and is defined by
(25)$$v_{k,i+1}=\lambda \left(\beta v_{k,i+1}^{a}\right)+\left(1-\lambda \right)v_{k,i}^{g}+\gamma \delta_{k,i}$$
where {λ,β,γ} are adjustable non-negative weighting coefficients. Algorithm 1 summarizes the methodology.
**Algorithm 1** Adaptive Cooperative Crowd Modeling using ATC**Require:** $w_{k,1}$, $t_{k,i}$, $L_{k,i}$ **for** i=1 to Number Iterations **do**    **for** k=1 to $N_{k}$ **do**      Adaptation step:      **if** $w_{k,i}\approx p_{k,i}$ **then**         $v_{k,i+1}^{a}=-C_{k,i}\left(\frac{R}{∥w_{k,i}-\left(p_{k,i}-\alpha \right)∥}-1\right)\left(\frac{w_{k,i}-\left(p_{k,i}-\alpha \right)}{∥w_{k,i}-\left(p_{k,i}-\alpha \right)∥}\right)$      **else**         $v_{k,i+1}^{a}=C_{k,i}\left(\frac{t_{k,i}-w_{k,i}}{∥t_{k,i}-w_{k,i}∥}\right)$      **end if**      $\delta_{k,i}=\frac{1}{|N_{k}|-1}\sum_{l\in N_{k}\backslash \left\{k\right\}}\left(∥w_{l,i}-w_{k,i}∥-r\right)u_{k,i}\left(w_{l,i}-w_{k,i}\right)$      $v_{k,i+1}=\lambda \left(\beta v_{k,i+1}^{a}\right)+\left(1-\lambda \right)v_{k,i}^{g}+\gamma \delta_{k,i}$      $\varphi_{l,i}=\left(1-\mu_{k}^{v}\right)v_{k,i-1}^{g}+\mu_{k}^{v}v_{k,i}$      Given $v_{k,i+1}$, we obtain the next location vector of node *k*      $w_{k,i+1}=w_{k,i}+\Delta i\cdot v_{k,i+1}$      Combination step:      $v_{k,i}^{g}=\sum_{l\in N_{k}}a_{l,k}^{v}\varphi_{l,i}$    **end for** **end for**

During each step of adaptation, people with disabilities can be advised on the direction and speed of his/her movement based on an estimate of the angle between movement direction and the direction towards the destination. This can be mathematically defined as
(26)$$\theta_{k,i}=\cos^{-1}\left(\frac{w_{k,i}-w_{k,i-1}}{∥w_{k,i}-w_{k,i-1}∥}\cdot \frac{t_{k,i}-w_{k,i}}{∥t_{k,i}-w_{k,i}∥}\right)$$
where ‘·’ refers to vector internal product and ∥·∥ refers to the Euclidean distance between two location vectors. $t_{k,i}$ is the target (or desired end point) of each node *k*, $w_{k,i-1}$ is the previous position of node *k*, and $w_{k,i}$ is its current position. $\theta_{k,i}$ can be delivered to the subjects with disabilities (or to their wheelchairs) to correct their directions where necessary.

The major advantage of diffusion adaptation compared with traditional consensus networks is its convexity, leading to stability of the multi-agent network [29].

### 2.2. Crowd Monitoring System

The proposed system can estimate and monitor the crowd movement and provide the necessary guidelines and warnings for any upcoming danger to the users, represented as agents of the same network, so they can navigate towards their destination while maintaining a safe social distance. For individuals with disabilities, the cooperative system can also assist them in moving through a crowd by providing them with the desired direction of movement and speed necessary to reach their desired destination (considered as the end point or target of the diffusion adaptation strategy).

In the proposed application, each agent, including the people with disabilities, carries a tracking device that can interact with its nearby tracking devices. Each tracking device receives the location information from all the nearby tracking devices. This information is fed to the diffusion adaptation model to be used as the nodes’ position at each time instant. Therefore, each tracking device acts as an intelligent node of the cooperative network.

In the scenario studied and presented here, for simulation purposes, all the tracking devices are Airtags from Apple Inc. Thanks to their Precision Finding feature, it is possible to locate other Airtag devices within a close proximity with a high precision within the centimeter range. Although other tracking devices are compatible with a wider range of OSs, such as the previously mentioned AirFinders or the latest Google or Samsung smartphones, these tracking devices are not always reliable, secure, and easily compatible with each other. On the contrary, Apple devices that come with embedded UWB high-precision location technology such as the iPhone 15, AirTag, or iWatch Series 9, can easily and securely connect to each other and provide a high-precision location of the devices.

In our experiment, the path geometry as well as the target location or end point ($t_{k,i}$) for our difussion adaptation model are presumed known. The agents are assumed to know the target beforehand, learn it through repetitions, follow the signs, or follow those who know them.

Each AirTag can be tracked by the nearby mobile devices and a mobile device can access the location of all the other tracking devices in the user’s neighborhood. This information is used as the location of each node *k* at each time instant *i* ($w_{k,i}$) for the model. A schematic diagram of the overall setup for a network of AirTag tracking devices is depicted in Figure 3. For visually impaired and blind users, the application is run in each individual’s smartphone and the recommendations and warnings are provided to the user through the speakers. In the case of wheelchair users, the application can be embedded within the wheelchair’s navigation control system, allowing the application to control the wheelchair movement based on the recommendations given by the diffusion adaptation algorithm.

In our experiment, each user needs to carry an AirTag and a mobile device, as it can be appreciated in Figure 3. Each AirTag is used as the transmitter to provide the agent’s location while each mobile device is used as the receiver to obtain the location of all the other agents of the network, represented in Figure 3 as all the AirTags connected to the agent’s mobile device.

To ensure the network is independent of the global network, the algorithm must be built-in and supported by a local communication protocol. However, here, we utilize the protocol provided by Apple to share the AirTag’s location with other Apple devices for simulation purposes. Therefore, each Apple device proximate to an AirTag sends its GPS location together with an encrypted message generated by the device, which can be considered as an identification message sent to the Apple Cloud. Then, the UWB technology of the device allows the devices to pin down the exact location of the other nearby tracking devices. The AirTags emit a beacon message constantly, which is picked up by the nearby iPhones, Macs, or other Apple devices. This allows the Apple Cloud to obtain the AirTag’s exact location in the centimeter range [30,31].

This cooperation between the iOS devices and the AirTags through UWB, as well as the other tracking devices, allows the creation of a reliable network able to locate the nearby devices with high precision even indoors. On the other hand, the available encryption system provides sufficient communication privacy. This is the main advantage of using Apple tracking devices compared to other devices.

## 3. Results

In this section, the crowd motion through a predefined path from a start to an end point is simulated. For a better evaluation of the proposed method, we simulate the crowd motion under a highly constrained situation. Therefore, the pathway chosen for the simulation presents a bottleneck situation, such as what the crowd encounters when passing through a narrow corridor as shown in Figure 4a,b, at the underground entrance to a castle as in Figure 4c, or a metro station corridor as in Figure 4d.

In the simulation, we consider the same target for all the nodes over time as the end point such as
(27)$$t_{k,i}\approx \hat{t}$$
where $\hat{t}$ represents the approximate target location, given that the agents keep their social distances even close to the target and, therefore, they do not converge exactly to a single target point.

The simulation parameters are set as follows. Consider a crowd of 40 people each representing a node of the network, where one node is a person with disabilities and the others are the general public. The step size, $\mu_{k}$, and $\mu_{k}^{v}$ are set to 0.05. For a safe distance from the walls, the coefficient α is set to 0.5. For velocity control, the coefficients {λ,β,γ} are, respectively, equal to {0.5,1,1}. The other parameters are set as defined in Section 2.1. For a more realistic representation, some random noise was added to the speed of the nodes as well as the distances between each other.

Figure 5 illustrates the movement of the crowd (mobile network) described above in ${ℜ}^{2}$. The green symbol ‘*’ on the right represents the end point (the target destination), the red dots represent the positions of the nodes considered as the general public, and the black dot represents the position of the people with disabilities over time. Finally, the blue lines define the walls of the path.

At the start of the simulation, in Figure 5a, the nodes are located at some random positions on the left side of the path. This represents the initial locations (start point) of each node showing the stage we start running the algorithm. These initial locations are generated randomly for simulation purposes. Later, all the nodes move towards their desired destination within the defined path. In Figure 5b,c, the nodes adapt their speeds and distances depending on the width of the path. Finally, in Figure 5d, which shows the end of the simulation time, the nodes gradually approach the desired destination.

In addition, Figure 6 provides the evidence that the nodes have effectively reached or will reach the target over time. This figure represents the Euclidean distances calculated using the following equation:(28)$$E_{k,i}={\left({\left(t_{x}-w_{{k,i}_{x}}\right)}^{2}-{\left(t_{y}-w_{{k,i}_{y}}\right)}^{2}\right)}^{1/2}$$
The Euclidean distance is calculated for each node *k* at each time instant *i*. Figure 6 shows that the average distance between all the nodes and the target decreases over time until it reaches a distance close to 0, which means that the nodes have reached the target. At the end of the simulation, some nodes have already reached their desired destination, while others, including those of the people with disabilities, are still approaching it. As explained in Section 2.1.3, the people with disabilities maintain a lower speed, so it will always take them longer to reach their destination than other nodes. This is also in line with the simulation presented in Figure 5, where it can be appreciated how people with disabilities fall behind the general public.

**Figure 6 bioengineering-11-00283-f006:**
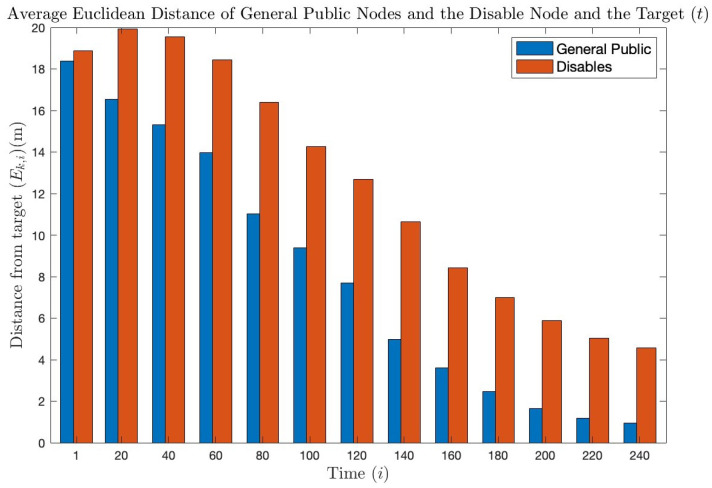
Representation of the average Euclidean distance between node *k* ($w_{k,i}$) and the target (t) at each time instant *i*. The blue bars represent the average distance of all the general public nodes, while the red bars represent the people with disabilities.

Table 1 shows how the chord length changes with respect to the path, and how the speed of the nodes is changed accordingly. The larger or smaller the chord, the lower or higher the speed of the node. The changes to the speed of the people with disabilities can also be appreciated in this table. For a similar chord length, the speed for node k=7, the general public, is considerably higher than the speed for node k=5, the people with disabilities.

In Table 2, it is evident that for a node representing the general public, the speed is higher and the distance is lower than for a node with a disability. This confirms the differences in the performance of the system for agents with different conditions.

These results show that the proposed method obtains a considerable improvement in accurate modeling of the movement of the crowd through a constrained path, compared to the results presented in [10,15]. In contrast to [15], the agents of the proposed method are able to keep a safe distance from the walls even in a bottleneck situation, and the special characteristics of the people with disabilities are implemented. On the other hand, in the proposed model, the distance and speed of all the nodes, especially the people with disabilities, are better regulated depending on the width of the path compared to the results presented in [10].

## 4. Conclusions

This paper explores the use of cooperative communication networks through diffusion adaptation together with the use of a suitable short-range communication technology such as UWB to model and monitor in real-time a moving crowd that can detect and avoid walls and other obstacles while moving towards a predefined target. The network nodes modify their speeds and distances based on the environmental geometrical properties and limitations. The simulation results show that the crowd successfully stays within the defined path, and the speed of each node as well as its distance from the nodes in its neighborhood are adapted to the new path profile and the predefined constraints. The proposed method is very impactful as it applies a high-end algorithm for decentralized cooperating networks to a real-life and very demanding problem such as monitoring and assisting people with disabilities in moving through a crowd. The proposed system also uses the state-of-the-art communication and positioning technology available in commercial devices to best safeguard the individuals, especially people with disabilities, in a highly challenging environment.

As mentioned in Section 1, previously proposed navigation assistive technology for people with disabilities, as well as recently proposed crowd monitoring systems, rely on image processing and computer vision [32,33]. These require an on-device high computational power and face high privacy concerns. Although our proposed system has some privacy concerns due to sharing the devices’ locations, this information is encrypted and stays within the network formed by the close-proximity tracking devices. On the other hand, existing crowd monitoring systems are centrally controlled and can only monitor a predefined area since they rely on the use of pre-installed surveillance cameras [34] or WiFi Beacons [35]. Finally, other proposed crowd modeling systems [36,37,38] can accurately simulate a crowd behavior under certain situations. However, compared to the proposed model, they cannot update the model in real-time based on the actual location of the agents or communicate the updated recommended direction or the speed of movement of the people with disabilities in underground situations or places with no internet connection.

Even so, the fully functional implementation of the proposed system in a real-world scenario still presents some challenges related to the use of Apple’s Precision Finding feature for the location of the nodes. Although, currently, iOS leads the mobile operating system market share in some countries, such as in the USA or Australia, Android remains the predominant mobile operating system worldwide [39]. This creates certain limitations on the correct implementation of the proposed system in certain countries or areas where a reliable network of Apple devices is not available. Although this could be alleviated with the use of more general UWB-embedded devices that do not need to be connected to a compatible mobile device, this approach also presents certain challenges. Although some standard protocols for UWB communications have been established, most UWB systems still present a high incompatibility with devices with different UWB radio chips [40].

Therefore, a major improvement in the application of the proposed cooperative system will be by integrating and embedding the system in each individual mobile phone that uses the same UWB standard protocol. By allowing the easy secured communication between UWB-embedded commercially available devices, we can obtain the relative coordinate of each node of the network relying only on short-range communication, which can eliminate the need for tracking devices connected to compatible mobile phones. The aim is to have the system be independent of long-range communication or tracking systems such as GPS. The devices can be empowered by the necessary embedded software enabling short-range communication networks using Bluetooth, UWB, or any other suitable low-range secured network. On the other hand, future work and improvements on other positioning systems for indoor and outdoor locations, such as in the feasibility of using iBeacon-based positioning systems [41] in open areas, can improve the system performance. 

## Figures and Tables

**Figure 1 bioengineering-11-00283-f001:**
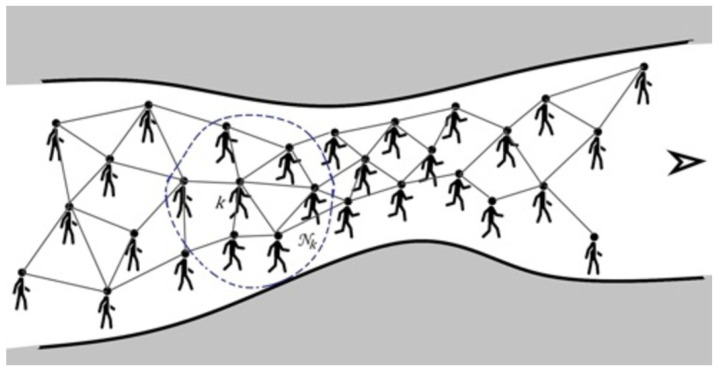
The network agents confined by two walls move in a geometrically varying environment whereby their speeds and their proximities can change accordingly. The neighborhood of agent *k*, which represents the agent with disabilities in the network, is denoted by $N_{k}$ represented by the dashed line. The crowd movement direction is represented by the arrow at the right of the path.

**Figure 2 bioengineering-11-00283-f002:**
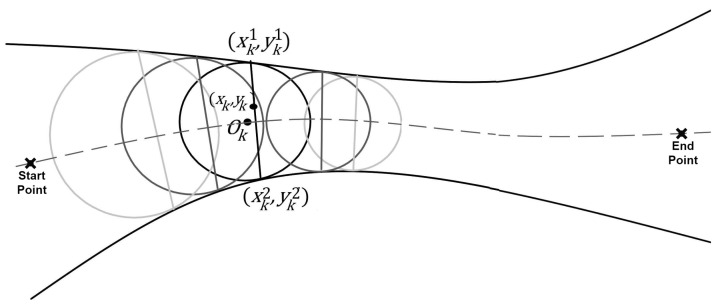
Using tangent circles to estimate the varying width of the space in ${ℜ}^{2}$ between the start and end points of the path for each node *k* in the coordinates ($x_{k},y_{k}$). The circle chord length (between the two tangent points) that contains node *k* best represents the width of the pathway.

**Figure 3 bioengineering-11-00283-f003:**
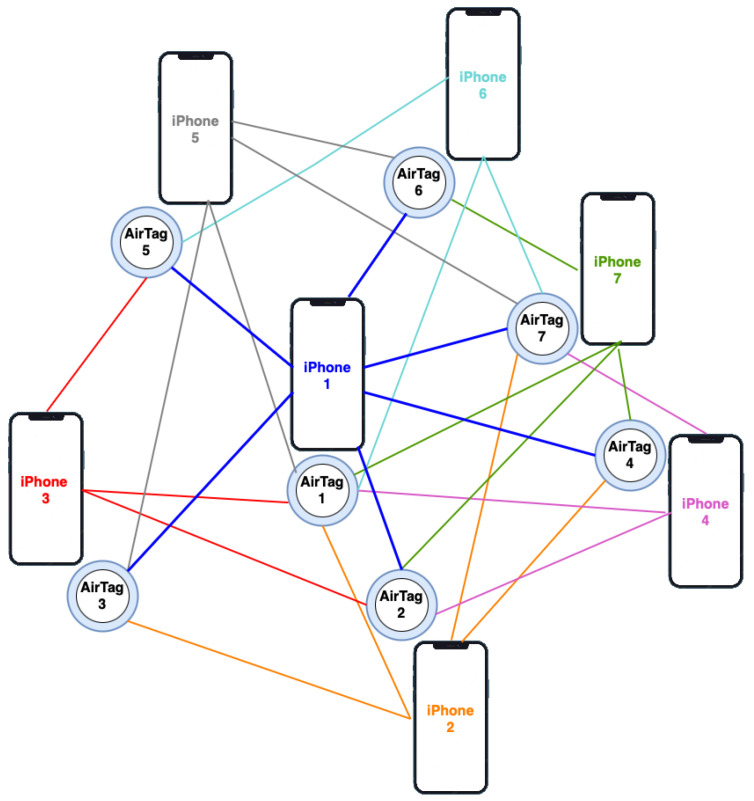
Illustration of the setup for using AirTag tracking devices as the agents of a cooperative network. Each AirTag is with an agent and can be tracked by the nearby iPhones. Each colored line represents the interaction and share of location for an agent, represented by a pair of iPhones and its AirTag, and other tracking devices within a close proximity, which forms the neighborhood $N_{k}$.

**Figure 4 bioengineering-11-00283-f004:**
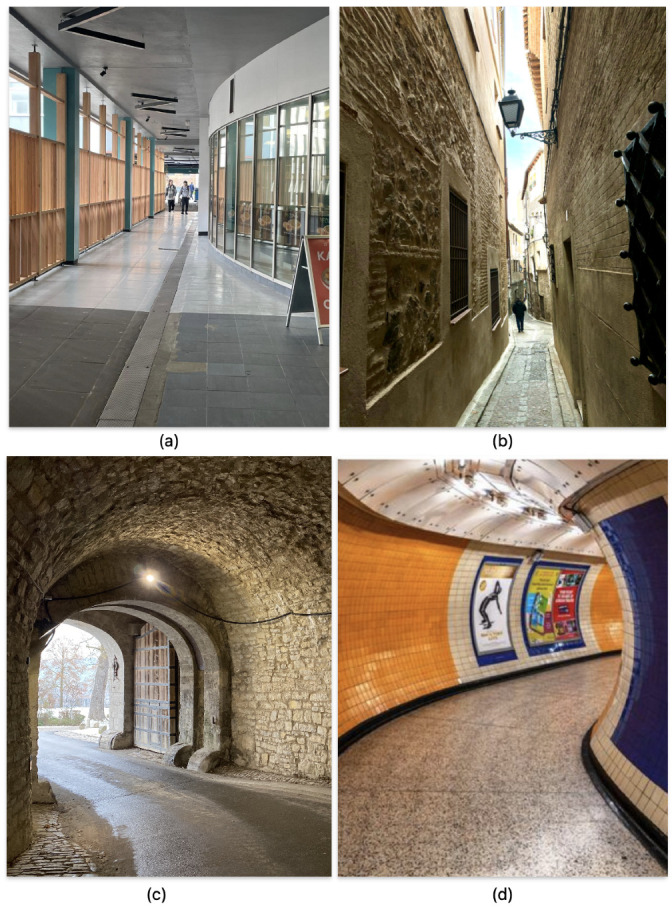
Illustration of pathways with possible bottleneck represented by the simulated pathway. (**a**) Narrowing of a passageway, (**b**) a pathway with a narrow corridor, (**c**) the underground entrance to a castle, and (**d**) the corridor of a metro station.

**Figure 5 bioengineering-11-00283-f005:**
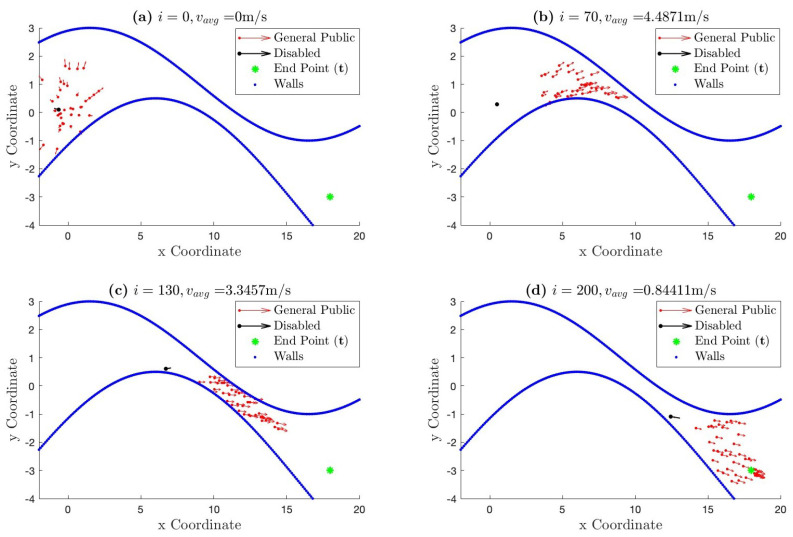
Simulation of the movement of a crowd over time. The average speed ($v_{k,i}$) of all the nodes is given for i=0, i=70, i=130, and i=200. The path is represented by blue dots, the general public by red dots, the people with disabilities by a black dot, and the end point of the path by a green “*”.

**Table 1 bioengineering-11-00283-t001:** Length of the chord ($L_{k,i}$) that goes through nodes k=7 and k=5 at several time instants *i* and speed ($v_{k,i}^{c}$) of nodes k=7 and k=5, respectively. Node k=5 represents the individual with disabilities.

	*i* = 2	*i* = 40	*i* = 80	*i* = 100	*i* = 150	*i* = 200
$L_{7,i}$ (m)	5.47	3.31	2.36	2.03	3.32	5.75
$v_{7,i}^{c}$ (m/s)	2.11	2.83	3.15	3.26	2.83	2.02
$L_{5,i}$ (m)	4.57	4.63	3.99	3.68	2.13	2.53
$v_{5,i}^{c}$ (m/s)	1.2	1.2	1.30	1.35	1.61	1.54

**Table 2 bioengineering-11-00283-t002:** Average distances between nodes k=7 and k=5 and their respective neighbors ($r_{k,i}$), and their speeds ($v_{k,i}$) at several time instants *i*. Node k=5 represents a node with a disability.

	*i* = 2	*i* = 12	*i* = 20
$r_{7,i}$ (m)	1.64	1.36	1.48
$v_{7,i}^{c}$ (m/s)	2.88	3.26	2.02
$r_{5,i}$ (m)	5.37	4.14	4.89
$v_{5,i}^{c}$ (m/s)	2.70	0.98	1.89

## Data Availability

The data presented in this study were generated using the source code openly available in https://github.com/AliciaFalconCaro/CrowdModelingSimulationWithDisables (accessed on 12 March 2024).

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
