# Peer review of "Adaptive Network Model for Assisting People with Disabilities through Crowd Monitoring and Control"

_bioengineering, 2024, doi:10.3390/bioengineering11030283_

Round 1

Reviewer 1 Report

Comments and Suggestions for Authors

The paper proposes a novel crowd modeling approach using diffusion adaptation that incorporates environmental parameters, an assistive technology based on diffusion adaptation, and utilizing smart devices for crowd monitoring.

The paper is nice and interesting; however, I have several concerns:

1. The authors explain what d_k,i is by using w_k,i; however, what is w_k,i?

2. In equations 3 and 4 the authors define Tau by Psi and Psi by Tau. This looks like a recursive setup. What is the stopping condition?

3. There is an equation between equation 13 and 14 that is not numbered.

4. In this unnumbered equation, why did the authors use the constant 3.6 and not another number? An explanation is needed.

5. There are two equations between equation 19 and 20 that are not numbered.

6. In Figure 5, where is the start point?

7. The authors discuss in the first section why GNSS/GPS is not good enough for their application. Today there are works like Rakhmanov A., "Compression of GNSS Data with the Aim of Speeding up Communication to Autonomous Vehicles", Remote Sensing, 2023, Vol. 15(8), paper no. 2165. Available online at: https://www.mdpi.com/2072-4292/15/8/2165 and also Correia, S.D., Perez, R., Matos-Carvalho, J., Leithardt, V.R.Q. μJSON, "A Lightweight Compression Scheme for Embedded GNSS Data Transmission on IoT Nodes", In IEEE 2022 5th Conference on Cloud and Internet of Things (CIoT), Piscataway, NJ, USA, 2022.  that explain how to lower the level of errors and make the GNSS/GPS devices more accurate. Probably in the near future even a standard GNSS/GPS device will be sufficient for the application proposed here. I would encourage the authors to cite these papers and explain that while they currently had to work with a different device, in the not too distant future when GNSS/GPS devices are improved, they will be able to use any GNSS/GPS and thus their application will be even more available.

8. The format of references should be consistent.

Reviewer 2 Report

Comments and Suggestions for Authors

Paper is an extended version of conference proceeding paper (Falcon-Caro, A., & Sanei, S. (2022, November). Cooperative Networking Approach to Assisting Blinds in a Crowd Using Air Trackers. In 2022 4th International Conference on Emerging Trends in Electrical, Electronic and Communications Engineering (ELECOM) (pp. 1-5). IEEE) and  introduces an interesting investigation by modeling the crowd movement and introduce a cooperative learning approach to enable cooperation and self-organization of the crowd members with impaired health or on wheelchair to ensure their safe movement in the crowd. The results are based on simulation of an open space (outdoor environment, with GPS signal available), however, the selection of the wall shape seems to be not justified and not very close to realistic environment. Since the model is created, it would be valuable to investigate, how alternative optimization algorithms are able to deal wit the presented task. The comparison with several perspective alternatives would draw more clear contribution not only to the fact that the simulation model is working correctly, but also would justify the selection of diffusion adaptation approach fo this task. The conclusions looks more as a statement of the performed work, it would be nice to have a comparison with an alternative approarch to illustrate numerically the efficiency of the selected approach in the ligth of other alternatives.

Here are additional details according to the given points:

1. What is the main question addressed by the research?

The main objective of the proposed solution to provide a feedbak for disable people to navigate them towards the desired destination avoiding collision with walls and other moving humankind on the path to destination point. The feedback is estimated by proposed algorithm, working with measurable imput parameters and performing simulation of the crowd behavior.

2. What parts do you consider original or relevant for the field? What specific gap in the field does the paper address? 

The original contribution in this paper is related to the proposed Algorithm, provided in the paper, which us based on equation of the effective social distance in the neighborhood of node k at time instant i and the agent based collaboration approach.

3. What does it add to the subject area compared with other published material?

The paper proposes to apply Diffusion adaptation modeling in modeling crowd behavior which seems not be used previously in this specific application area.

4. What specific improvements should the authors consider regarding the methodology? What further controls should be considered?

The improvements in methodology vcould be made by providing additional insights in stability of GPS based systems, providing accurate position. It is not considered in the presented research.

5. Please describe how the conclusions are or are not consistent with the evidence and arguments presented. Please also indicate if all main questions posed were addressed and by which specific experiments.

The conclusions should show the advantages of proposed solution comparing it to some previously presented crowd behavior modeling approaches, such as:

- Ying, Liu., Ying, Liu., Cheng, Sun., Yiming, Bie. (2015). Modeling Unidirectional Pedestrian Movement: An Investigation of Diffusion Behavior in the Built Environment. Mathematical Problems in Engineering, 2015:1-6. doi: 10.1155/2015/308261

- Sujeong, Kim., Stephen, J., Guy., Dinesh, Manocha., Ming, C., Lin. (2012). Interactive simulation of dynamic crowd behaviors using general adaptation syndrome theory.  55-62. doi: 10.1145/2159616.2159626

Or other alternatives that authors find competetive to their approaches.

6. Are the references appropriate?

Yes. References includes some review papers, papers related to crowd behavior feature extraction, application of Air Tags, successful application of Diffusion Adaptation Approach to solve problems in other application areas.

7. Please include any additional comments on the tables and figures and  quality of the data.

Figures and tables provides valuable insigts, illustrating dynamics and distance changes of the nodes over the time. Since the speed of node movement is defined, it would be possible to add time measurement units to the presented time offsets it Table 1, Table 2, Figure 6. 

Reviewer 3 Report

Comments and Suggestions for Authors

The following aspects must be improved:

-not clearly what is exact the scope of the paper: please explain more clearly

-add a related work section with methods (and their performances) that are similar (in scope) with the proposed method

-extend the result section by comparing obtained results with other existing ones

-regarding the results section: what about real life scenarios - what about the processing time?  Are there any overheads that can affect real life scenarios?

Round 2

Reviewer 1 Report

Comments and Suggestions for Authors

The authors made a decent effort and the paper is certainly publishable so I would recommend accepting the paper.

Reviewer 3 Report

Comments and Suggestions for Authors

Since all my comments were addressed, I recommend to publish the paper.